# Multiscale Information Decomposition Dissects Control Mechanisms of Heart Rate Variability at Rest and During Physiological Stress

**DOI:** 10.3390/e21050526

**Published:** 2019-05-24

**Authors:** Jana Krohova, Luca Faes, Barbora Czippelova, Zuzana Turianikova, Nikoleta Mazgutova, Riccardo Pernice, Alessandro Busacca, Daniele Marinazzo, Sebastiano Stramaglia, Michal Javorka

**Affiliations:** 1Biomedical Center Martin, Jessenius Faculty of Medicine in Martin, Comenius University in Bratislava, 03601 Martin, Slovakia; 2Department of Physiology, Jessenius Faculty of Medicine in Martin, Comenius University in Bratislava, 03601 Martin, Slovakia; 3Department of Engineering, University of Palermo, 90128 Palermo, Italy; 4Data Analysis Department, Ghent University, 9000 Ghent, Belgium; 5Dipartimento di Fisica, Universitá degli Studi Aldo Moro, 70126 Bari, Italy; 6Istituto Nazionale di Fisica Nucleare, 70126 Sezione di Bari, Italy

**Keywords:** heart rate variability, information decomposition, multiscale analysis, redundancy, synergy

## Abstract

Heart rate variability (HRV; variability of the RR interval of the electrocardiogram) results from the activity of several coexisting control mechanisms, which involve the influence of respiration (RESP) and systolic blood pressure (SBP) oscillations operating across multiple temporal scales and changing in different physiological states. In this study, multiscale information decomposition is used to dissect the physiological mechanisms related to the genesis of HRV in 78 young volunteers monitored at rest and during postural and mental stress evoked by head-up tilt (HUT) and mental arithmetics (MA). After representing RR, RESP and SBP at different time scales through a recently proposed method based on multivariate state space models, the joint information transfer $T_{\mathrm{RESP},\mathrm{SBP}\to \mathrm{RR}}$ is decomposed into unique, redundant and synergistic components, describing the strength of baroreflex modulation independent of respiration ($U_{\mathrm{SBP}\to \mathrm{RR}}$), nonbaroreflex ($U_{\mathrm{RESP}\to \mathrm{RR}}$) and baroreflex-mediated ($R_{\mathrm{RESP},\mathrm{SBP}\to \mathrm{RR}}$) respiratory influences, and simultaneous presence of baroreflex and nonbaroreflex respiratory influences ($S_{\mathrm{RESP},\mathrm{SBP}\to \mathrm{RR}}$), respectively. We find that fast (short time scale) HRV oscillations—respiratory sinus arrhythmia—originate from the coexistence of baroreflex and nonbaroreflex (central) mechanisms at rest, with a stronger baroreflex involvement during HUT. Focusing on slower HRV oscillations, the baroreflex origin is dominant and MA leads to its higher involvement. Respiration influences independent on baroreflex are present at long time scales, and are enhanced during HUT.

## 1. Introduction

Short-term changes in heart rate (HR)—heart rate variability (HRV)—are the result of the activity of several mechanisms, which operate across multiple temporal scales ranging from seconds to minutes [1,2,3]. Oscillations at short time scales represented by the activity in the high-frequency band (HF, 0.15–0.40 Hz), correspond to HR oscillations related to the respiration, i.e., to respiratory sinus arrhythmia (RSA). Low-frequency band HR oscillations (LF, 0.04–0.15 Hz) result mostly from blood pressure oscillations reflecting vasomotor activity (Mayer waves) transferred to HR through the baroreflex [1,2,4,5,6]. Longer time scales included in very low frequency oscillations (VLF, 0.0033–0.04 Hz) are also mediated by thermoregulation and the hormonal control (e.g., renin–angiotensin–aldosterone system) influencing vasomotion and heart rate [2,4,6]. The HRV signal resulting from these mechanisms deployed across multiple time scales is widely used and has been studied for many years as a noninvasive marker of cardiovascular autonomic control [1,7]. Moreover, it became more important when it was realized that indices derived from HRV are highly sensitive in determining a dysfunction of the cardiovascular control system occurring as a consequence, but also as a pathomechanism, of many pathological states [8,9,10,11,12,13,14].

In the last decades, a variety of different approaches have been proposed for the analysis of HRV, ranging from time domain approaches to frequency domain analysis and information-theoretic methods. At the same time, HRV was analysed in conjunction with the oscillatory activity of other physiological signals to improve the understanding and interpretability of the cardiac rhythms, e.g., when they are assessed in the context of cardiovascular and cardiorespiratory control. Indeed, HRV (usually evaluated in terms of beat-to-beat heart period changes represented by the RR interval) was mainly studied in bivariate analysis together with the variations in systolic blood pressure (SBP) [15,16,17,18,19] or in the respiration pattern (RESP) [20,21]. Later, the analysis has been extended to a multivariate setting, where cardiovascular and cardiorespiratory oscillations were evaluated simultaneously, which helped to understand the network of interconnections among variables, shedding light on the combined activity of physiological mechanisms like the baroreflex and the RSA [22,23,24,25].

HRV mechanisms differ in relation to the time scale at which it is observed, which is related to the different frequency of HR oscillations. Since coupling and causality can be specific for a given time scale of oscillatory activity, it would be of benefit to better characterise the mechanisms and interactions governing HRV on various time scales [23]. However, the analysis of descriptive measures of causality such as the information transfer is currently focused mostly on the “raw” measured time series (containing VLF, LF and HF oscillations) [26,27], for which causality indices are predominantly influenced by the time scales encompassing the most pronounced oscillatory activity, masking effects at the other time scales. Therefore, a separate analysis of the interactions with different time scales could provide information that is not detectable by the information-theoretic method applied so far in previous studies [26].

Recent developments in the field of information dynamics can be relevant to improve our understanding on how the short-term HRV interacts with the cardiovascular and respiratory oscillations across various time scales. Indeed, tools of information decomposition [28] allow to assess separately the contribution that different sources of information bring to the dynamics of an assigned target variable, as well as to investigate the nature of the interaction between sources sending information to the target. In this study we exploit the so-called multiscale partial information decomposition [29] in order to quantify the amount of information transferred from SBP and RESP towards HR (represented by its reciprocal value—RR interval), and also to identify the type of interaction (synergistic or redundant) between the SBP and RESP while they transfer information to RR interval oscillations across multiple temporal scales. According to the theory of information decomposition, two sources are redundant if each carries individually information about the same aspects of the target, while synergistic information can only be retrieved combining correlated sources activity.

While this type of analysis was previously applied only in a preliminary study analysing patients undergoing coronary artery bypass graft [30], an exhaustive evaluation of how information is transferred across different time scales from SBP and RESP towards RR interval oscillations in different physiological states is still missing. Therefore, in the present study we perform multiscale information decomposition of cardiovascular and cardiorespiratory interactions in healthy subjects monitored in a relaxed physiological condition (i.e., supine rest) and during two conditions inducing different types of physiological stress (i.e., postural and mental stress). The main aim of the study is to assess noninvasively, from short-term SBP, RESP and RR oscillations, the two major physiological mechanisms underlying HRV (i.e., the RSA and the high-pressure baroreflex), exploiting an information-theoretic interpretation of these mechanisms to infer their degree of involvement and mutual interaction at rest and in response to different stressors.

## 2. Materials and Methods 

### 2.1. Experimental Protocol

In the present study, a group of healthy young volunteers (78 subjects; 32 m/46 f; age range: 16.0–25.8 years; median age: 18.7 years) underwent ECG (horizontal bipolar thoracic lead; CardioFax ECG-9620, NihonKohden, Japan), continuous finger arterial blood pressure (Finometer Pro, FMS, Netherlands) and respiratory volume signal (respiratory inductive plethysmography; RespiTrace, NIMS, USA) recording during supine rest (15 min), head-up tilt (HUT, the subject was tilted to 45 degrees on the motor driven tilt table for 8 min to evoke mild orthostatic stress), supine recovery (10 min) and mental arithmetic task (MA) in the supine position (6 min). For more detailed information about the protocol see Javorka et al. [31]. Participants were instructed not to use substances influencing autonomic nervous system or cardiovascular system activity during 24 h before the measurement. Female volunteers were examined in the proliferative phase (6th–13th day) of their menstrual cycle. The study was approved by Ethical Committee of the Jessenius Faculty of Medicine, Comenius University and all participants signed a written informed consent.

### 2.2. Time Series Extraction

All measured signals were digitised at 1000 Hz. The following beat-to-beat time series were measured from the acquired signals: the RR interval was calculated as the temporal distance between two consecutive R waves (RR_n_); the SBP was taken as the maximum value of the arterial blood pressure inside the n-th RR interval and was denoted as SBP_n_; and the respiration volume signal (RESP_n_) was sampled at the onset of the first R-wave peak delimiting RR_n_. For each subject, sequences of N=300 values were extracted from the original recordings for data analysis for each phase of the protocol separately (see Figure 1). To avoid transient changes, the sequence of 300 beats for supine rest phase started 8 min after the beginning of the supine rest phase, for HUT 3 min after the change of the position, for supine recovery 7 min before starting the MA task and for MA 2 min after starting of the MA task. During the whole protocol the respiration frequency for each volunteer was > 0.15 Hz.

### 2.3. Multiscale Information Decomposition

In order to quantify the amount of information transferred from SBP and RESP (sources) to the RR (target) across multiple temporal scales we applied the method described in [29,30], which exploits recent developments in information-theory—the partial information decomposition (PID) introduced by Williams and Beer [32]—and applies them to the study of information dynamics [28]. The PID approach allows decomposition of the information shared between a target process and two source processes into unique, redundant and synergistic contributions. In particular, while the traditional information decomposition method (i.e., interaction information decomposition (IID) [28]) provides a single measure quantifying the net balance between redundancy and synergy, the PID accounts for the fact that synergy and redundancy may coexist as distinctive elements of the information transferred by multiple sources towards a target and allows quantification of these two elements separately [32].

Traditionally, in an information-theoretic framework the directed transfer of information between components of a network of interacting processes is assessed by the transfer entropy (TE). The TE from a source process to a target process quantifies the amount of information contained in the past states of the source that can be used to predict the present state of the target beyond the information contained in its own past. In our case, we consider the RR interval as the target process and the SBP and RESP as the source processes. The information transferred individually from SBP to RR and from RESP to RR (individual TE) is then defined as
(1)$$T_{\mathrm{SBP}\to \mathrm{RR}}=I\left({\mathrm{RR}}_{n};{\mathrm{SBP}}_{n}^{-}{|\mathrm{RR}}_{n}^{-}\right),$$
(2)$$T_{\mathrm{RESP}\to \mathrm{RR}}=I\left({\mathrm{RR}}_{n};{\mathrm{RESP}}_{n}^{-}{|\mathrm{RR}}_{n}^{-}\right),$$
where I(:;:|:) denotes conditional mutual information, ${\mathrm{RR}}_{n}$ denotes the present state of RR, and ${\mathrm{RR}}_{n}^{-}$, ${\mathrm{SBP}}_{n}^{-}$ and ${\mathrm{RESP}}_{n}^{-}$ represent the past states of RR, SBP and RESP, respectively. When the two sources are considered together, the information transferred toward RR from SBP and RESP (joint TE) is defined as
(3)$$T_{\mathrm{RESP},\mathrm{SBP}\to \mathrm{RR}}=I\left({\mathrm{RR}}_{n};{\mathrm{SBP}}_{n}^{-},{\mathrm{RESP}}_{n}^{-}{|\mathrm{RR}}_{n}^{-}\right).$$

In general, the joint TE differs from the sum of the two individual TEs. This happens because RESP and SBP typically interact with each other while they transfer information to RR. Such an interaction is synergistic if the two sources transfer more information to the target when they are considered together than when they are considered individually, and is redundant in the opposite case. The type of interaction is represented by the sign of the so-called interaction transfer entropy defined as [28]
(4)$$I_{\mathrm{RESP},\mathrm{SBP}\to \mathrm{RR}}=T_{\mathrm{RESP},\mathrm{SBP}\to \mathrm{RR}}-T_{\mathrm{SBP}\to \mathrm{RR}}-T_{\mathrm{RESP}\to \mathrm{RR}}.$$
Positive values of the interaction TE $I_{\mathrm{RESP},\mathrm{SBP}\to \mathrm{RR}}$ denote synergy, where the joint TE $T_{\mathrm{RESP},\mathrm{SBP}\to \mathrm{RR}}$ is greater than the sum of the two individual TEs $T_{\mathrm{SBP}\to \mathrm{RR}}\;\mathrm{and}\;T_{\mathrm{RESP}\to \mathrm{RR}}$. In contrast, negative values of the interaction TE refer to redundancy, which occurs when the information from the sources are overlapped, meaning that the sum of individual TEs is larger than the joint TE. The main disadvantage of this decomposition of the joint TE (denoted as IID, see Figure 2a) is that the interaction TE is quantified using only one measure and thus makes redundancy and synergy mutually exclusive.

This deficiency can be overcome by the PID. Using this decomposition one can express the joint TE $T_{\mathrm{RESP},\mathrm{SBP}\to \mathrm{RR}}$ as the sum of four separate terms: the unique TEs indicating unique contributions from RESP or from SBP to RR ($U_{\mathrm{RESP}\to \mathrm{RR}}$ and $U_{\mathrm{SBP}\to \mathrm{RR}}$, respectively), the redundant TE ($R_{\mathrm{RESP},\mathrm{SBP}\to \mathrm{RR}}$) and the synergistic TE ($S_{\mathrm{RESP},\mathrm{SBP}\to \mathrm{RR}}$). These four terms results as non-negative components of the joint information transferred by the two sources RESP and SBP to the target RR (Figure 2b). As shown in Figure 2b, the PID is defined in a way such that the unique TE from one source to the target is obtained removing from the individual TE the amount of redundant information shared with the other source:(5)$$U_{\mathrm{SBP}\to \mathrm{RR}}=T_{\mathrm{SBP}\to \mathrm{RR}}-R_{\mathrm{RESP},\mathrm{SBP}\to \mathrm{RR}},$$
(6)$$U_{\mathrm{RESP}\to \mathrm{RR}}=T_{\mathrm{RESP}\to \mathrm{RR}}-R_{\mathrm{RESP},\mathrm{SBP}\to \mathrm{RR}}.$$
Moreover, substituting (5) and (6) in (4) and recalling that the PID can be expressed as $T_{\mathrm{RESP},\mathrm{SBP}\to \mathrm{RR}}=U_{\mathrm{SBP}\to \mathrm{RR}}+U_{\mathrm{RESP}\to \mathrm{RR}}$+$R_{\mathrm{RESP},\mathrm{SBP}\to \mathrm{RR}}+S_{\mathrm{RESP},\mathrm{SBP}\to \mathrm{RR}}$, the synergistic TE can be obtained as
(7)$$S_{\mathrm{RESP},\mathrm{SBP}\to \mathrm{RR}}=I_{\mathrm{RESP},\mathrm{SBP}\to \mathrm{RR}}+R_{\mathrm{RESP},\mathrm{SBP}\to \mathrm{RR}}.$$
Equations (5)–(7) show that the terms composing the PID can be derived from known information theoretic measures like the joint and individual TE and from the redundancy $R_{\mathrm{RESP},\mathrm{SBP}\to \mathrm{RR}}$. Therefore, to complete the PID a definition for the redundant information is needed, and to this end a growing body of research is in progress [33]. Here we make reference to the so-called minimum mutual information PID (MMI-PID) [34], according to which redundancy is simply defined as the minimum information transferred individually from each source to the target:(8)$$R_{\mathrm{RESP},\mathrm{SBP}\to \mathrm{RR}}=\min\{T_{\mathrm{RESP}\to \mathrm{RR}},T_{\mathrm{SBP}\to \mathrm{RR}}\}.$$
This definition, which completes the PID making all terms computable, has the desirable properties that the redundant TE is independent on the correlation between the sources and that many other definitions of redundancy converge to this one for linear Gaussian multivariate processes [34].

The measures of information transfer defined above were computed across different time scales using the approach described in [29,30]. This approach describes the observed multivariate process as a vector autoregressive (VAR) process, and provides mathematical derivations of the VAR parameters as a function of the time scale. In this way, all terms appearing in the IID and PID decompositions are computed analytically from the prediction error variances of the VAR model represented at any assigned time scale τ. The method makes use of state space models to represent the VAR parameters after rescaling the original process at scale τ; rescaling consists of the two subsequent steps of filtering (performed using a low-pass filter with cut-off frequency 1/2τ), and then downsampling (taking one every τ samples) the original process. This overall approach brings the advantage that the rescaling procedure is not directly applied to the series, but its effect is extrapolated from the theoretical evaluation of the state space parameters after the filtering and downsampling steps. We refer to Reference [29,30] for full details about implementation of the multiscale approach.

### 2.4. Data Analysis and Interpretation

The multiscale method described in Section 2.3 allows for computing all measures of information decomposition, at any assigned time scale, τ, directly from the coefficients of the VAR process fitting the original time series. In this study, VAR model identification was performed for each set of time series representing the multivariate process (RESP, SBP, RR) using the least squares method, and setting the model order according to the Akaike Information Criterion [35]. The model was extended to incorporate zero-lag effects from RESP to SBP and to RR, and from SBP to RR, thus allowing fast effects of respiration on cardiovascular variables and within-beat baroreflex influences [28].

The method was applied first computing the joint, interaction, unique, redundant and synergistic information transfers for time scales τ ranging from 1 to 12. Then, a scale-specific analysis was performed to assess information transfer separately with regard to oscillatory components with meaningful physiological interpretation contained in the time series. Specifically, in order to distinguish effects due to all the oscillations from effects due to slower oscillations only (which represent the oscillations in VLF and LF band—VLF + LF) we computed the information measures at two time scales: $\tau_{1}=1$, corresponding to raw non-rescaled data, and $\tau_{2}$ determined—for each subject and experimental condition—as the time scale which removes the HF band using the formula
(9)$$\tau_{2}=\frac{1}{2f_{\tau }\;\bar{\mathrm{RR}}},$$
where $f_{\tau }$ is the cut-off frequency of the low-pass filter used for rescaling and is equal to 0.15 Hz, and $\bar{\mathrm{RR}}$ is the mean RR interval (measured in seconds).

With the above analysis we obtain, at the time scales including all oscillations ($\tau_{1}$) or removing HF oscillations ($\tau_{2}$), estimates of all measures of information transfer resulting from IID and PID. In order to ease physiological interpretation of these measures, in the following we discuss the meaning of the PID components based on (i) expected physiological mechanisms underlying cardiovascular and cardiorespiratory influences and (ii) methodological derivations that relate the PID and IID decomposition. Our physiological assumptions, depicted in Figure 2c, are that respiration acts as an exogenous input on the two cardiovascular variables (i.e., RESP affects SBP and RR without being substantially affected by them); moreover, we consider two possible mechanisms whereby respiration influences the variability of heart rate [36,37]: baroreflex-mediated respiratory effects on heart rate oscillations (indirect pathway RESP→SBP→RR) and direct effects unrelated to SPB (possibly of central origin; a direct pathway RESP→RR). In addition, an influence of slower SBP and respiratory pattern oscillations (e.g., of vasomotor origin or respiratory pattern variability-related, respectively) on RR transferred via baroreflex are also considered on longer time scales ($\tau_{2}$).

Methodologically, as demonstrated in Appendix A, the MMI PID is such that one source (the one providing the lowest amount of information about the target) contributes to the target dynamics with no unique information, but interacting with the other source to provide an amount of redundant (shared) information equivalent to its individual information transfer and an amount of synergistic (complementary) information equivalent to its conditional information transfer. This means that, if $T_{\mathrm{RESP}\to \mathrm{RR}}$<$T_{\mathrm{SBP}\to \mathrm{RR}}$, the MMI PID will yield
(10)$$\begin{array}{c} U_{\mathrm{RESP}\to \mathrm{RR}}=0,\\ R_{\mathrm{RESP},\mathrm{SBP}\to \mathrm{RR}}=T_{\mathrm{RESP}\to \mathrm{RR}},\\ S_{\mathrm{RESP},\mathrm{SBP}\to \mathrm{RR}}=T_{\mathrm{RESP}\to \mathrm{RR}|\mathrm{SBP}},\\ U_{\mathrm{SBP}\to \mathrm{RR}}=T_{\mathrm{SBP}\to \mathrm{RR}}-T_{\mathrm{RESP}\to \mathrm{RR}}; \end{array}$$

If, on the contrary, if $T_{\mathrm{SBP}\to \mathrm{RR}}$ < $T_{\mathrm{RESP}\to \mathrm{RR}}$, the MMI PID will yield
(11)$$\begin{array}{c} U_{\mathrm{SBP}\to \mathrm{RR}}=0,\\ R_{\mathrm{RESP},\mathrm{SBP}\to \mathrm{RR}}=T_{\mathrm{SBP}\to \mathrm{RR}},\\ S_{\mathrm{RESP},\mathrm{SBP}\to \mathrm{RR}}=T_{\mathrm{SBP}\to \mathrm{RR}|\mathrm{RESP}},\\ U_{\mathrm{RESP}\to \mathrm{RR}}=T_{\mathrm{RESP}\to \mathrm{RR}}-T_{\mathrm{SBP}\to \mathrm{RR}}. \end{array}$$

In both cases, redundancy $R_{\mathrm{RESP},\mathrm{SBP}\to \mathrm{RR}}$ relates to the common information shared by the sources (RESP, SBP) about the target RR, and is in our context associated with the information transferred along the pathway RESP→SBP→RR describing baroreflex-mediated respiratory effects on heart rate (red + green arrows in Figure 2c). The amounts of unique information transfer are relevant to information flowing from one source to the target without involving the other source, associated here to respiration-unrelated baroreflex effects ($U_{\mathrm{SBP}\to \mathrm{RR}}$, reflecting the path SBP→RR; green arrow in Figure 2c) or to the nonbaroreflex mechanism of RSA ($U_{\mathrm{RESP}\to \mathrm{RR}}$, reflecting the path RESP→RR; blue arrow in Figure 2c). Finally, synergy $S_{\mathrm{RESP},\mathrm{SBP}\to \mathrm{RR}}$ is the information that the source sending no unique information provides about the target after conditioning on the other source (i.e., $T_{\mathrm{RESP}\to \mathrm{RR}|\mathrm{SBP}}$ when $U_{\mathrm{RESP}\to \mathrm{RR}}=$0, and $T_{\mathrm{SBP}\to \mathrm{RR}|\mathrm{RESP}}$ when $U_{\mathrm{SBP}\to \mathrm{RR}}=0$; this suggests that $S_{\mathrm{RESP},\mathrm{SBP}\to \mathrm{RR}}>0$ corresponds to $T_{\mathrm{RESP}\to \mathrm{RR}|\mathrm{SBP}}>0$ when $U_{\mathrm{SBP}\to \mathrm{RR}}>0$, and to $T_{\mathrm{SBP}\to \mathrm{RR}|\mathrm{RESP}}>0$ when $U_{\mathrm{RESP}\to \mathrm{RR}}>0$). Given this, synergy quantifies the contemporaneous presence of information transfer along both the pathways whereby the sources can affect the target (with unique transfer from one source and conditional transfer from the other source); therefore, we associate synergy between RESP and SBP with the simultaneous involvement of both pathways whereby respiration affects the heart rate (i.e., RESP→RR and RESP→SBP→RR; blue and red + green arrows in Figure 2c).

### 2.5. Statistical Analysis

Due to the non-Gaussian distribution of the information measures, the statistical comparison of a given measure across conditions (supine rest, HUT; supine recovery, MA) was performed using the nonparametric Friedman test. If the Friedman test indicated statistical significance, the Conover test for pairwise comparisons was used as a post hoc test; the test corrects for multiple comparisons, and here was implemented considering two post-hoc pairwise comparisons: supine rest vs. HUT, and supine recovery vs. MA. The comparison between the raw data (scale $\tau_{1}=1$) and slower oscillations (scale $\tau_{2}$) within the same phase was performed using the Wilcoxon signed-rank test. All results were considered statistically significant at a *p*-value < 0.05. Effect sizes were quantified by Kendall’s coefficient of concordance W. The statistical analysis was performed using SYSTAT 13 (Systat Software Inc., USA). Full results of the statistical analysis are reported in Appendix B.

## 3. Results

Figure 3 shows the results of multiscale information decomposition representing the trends across multiple time scales of the analysed TE components: joint TE ($T_{\mathrm{RESP},\mathrm{SBP}\to \mathrm{RR}}$), interaction TE ($I_{\mathrm{RESP},\mathrm{SBP}\to \mathrm{RR}}$), unique TE ($U_{\mathrm{RESP}\to \mathrm{RR}},U_{\mathrm{SBP}\to \mathrm{RR}}$), redundant TE ($R_{\mathrm{RESP},\mathrm{SBP}\to \mathrm{RR}}$) and synergistic TE ($S_{\mathrm{RESP},\mathrm{SBP}\to \mathrm{RR}}$) during supine rest, HUT, supine recovery and MA.

The joint TE was significantly higher during HUT compared to the preceding phase of supine rest for τ ≤ 7 (*p* ≤ 0.021), while during MA it was significantly lower compared to the preceding phase of supine recovery for 6 ≤ τ ≤ 10 (*p* ≤ 0.016) (Figure 3a). Compared to rest, the interaction information was significantly lower during HUT at scale 1 (*p* < 0.001), but it was significantly higher (*p* ≤ 0.046) from scale 3 to scale 5 (Figure 3b); in the transition from supine recovery to MA, the interaction information increased significantly at scale 6, 9, 10 and 12 (*p* ≤ 0.041). The unique TE from SBP to RR was significantly higher during HUT and MA at scale 1 (*p* < 0.001); statistical significance was reached also at scale 3 (*p* = 0.021) during HUT where significantly lower values in comparison with supine rest were found (Figure 3c).

A significant decrease of U_SBP→RR_ was observed also during MA over the last six scales (from 7 to 12, *p* ≤ 0.042). Moving from rest to HUT, the unique TE from RESP to RR decreased significantly at scale 1 (*p* < 0.001) and increased significantly at scales 3–6 (*p* ≤ 0.019) and 10–12 (*p* ≤ 0.048); during MA significantly lower values were found only at scale 1 (*p* = 0.031) (Figure 3d). The redundant TE from RESP, SBP to RR was higher at low scales (from 1 to 4, *p* < 0.001) and intermediate scales (from 5 to 7, *p* ≤ 0.014); during MA the significantly lower values were reached at scales 2, 8, and 9 (*p* ≤ 0.022) (Figure 3d). Finally, the synergistic TE was significantly higher during HUT almost across all time scales (from 1 to 9, *p* ≤ 0.008), while during MA no significant changes were found (0.949 ≥ *p* ≥ 0.141) (Figure 3f).

As a next step, in order to distinguish effects related to all oscillations (raw data — τ_1_ = 1) from effects due to slower oscillations only (VLF and LF band), we calculated the time scale individually for each recording corresponding to slower oscillations (τ_2_). The time scale τ_2_ was calculated using Equation (9) and rounded to the smallest possible integer value that was greater than or equal to the given number. The median values across subjects obtained for τ_2_ in the four phases of the experimental protocol were τ_2_ = 4 (range: 3–5) during supine rest, τ_2_ = 5 (range: 4–7) during HUT, τ_2_ = 4 (range: 3–6) during supine recovery, and τ_2_ = 5 (range: 3–7) during MA. In the following, results are presented with reference to Figure 4 showing each information measure separately for the time scales τ_1_ and τ_2_.

Applying IID analysis, the joint TE (Figure 4a) was significantly higher during HUT compared to the preceding rest phase for both scales (*p* < 0.001), while during MA it was significantly lower compared to preceding supine recovery only for slower oscillations (*p* = 0.019). The joint TE was significantly higher, in each of the four phases, when calculated for the raw data than for slower oscillations only (*p* < 0.001). The interaction information (Figure 4b) was significantly lower during HUT than during the preceding supine rest in the raw data (*p* < 0.001); moreover it was significantly higher in each of the four phases, when computed for the slower oscilations than for the raw data (*p* ≤ 0.010).

Using PID analysis, the unique TE from SBP to RR (Figure 4c) computed for the raw data was significantly higher during HUT and MA compared to the preceding resting phases (*p* < 0.001 and *p* = 0.002, respectively). No significant differences between phases were found in the unique TE from SBP to RR assessed for the slower oscilations (*p* ≥ 0.129); this measure was significantly higher for slower oscillations compared to raw data at rest (*p* < 0.001). The unique TE from RESP to RR (Figure 4d) computed for the raw data was significantly lower during HUT and MA compared to the preceding resting phases (*p* < 0.001 and *p* = 0.031, respectively), and was significantly lower for the slower oscillations in all phases (*p* < 0.001) except HUT (*p* = 0.174). The redundant and synergistic TE from RESP, SBP to RR (Figure 4e,f) were significantly higher during HUT compared to preceding supine rest in both raw data and slower oscilations (*p* ≤ 0.017). Redundancy and synergy were significantly lower for slower oscillations during the whole protocol (*p* < 0.001).

For both raw data and slow oscillations, significantly higher levels of redundancy in comparison with synergy were observed during the whole protocol (*p* ≤ 0.002). The only exception was a nonsignificant difference between synergy and redundancy levels for slower oscilations during HUT (*p* = 0.794).

## 4. Discussion

In the present study, we have faced the analysis of dynamic cardiovascular and cardiorespiratory interactions employing multiscale information decomposition to quantify the amount of information transferred from respiration and arterial pressure variability, considered as source processes, to heart rate variability, considered as target process. Our findings illustrate the contribution of baroreflex and nonbaroreflex mechanisms to respiration-related HRV (RSA) [37], and the contribution of baroreflex and respiratory mechanisms across different time scales, elucidating the origin of HRV in resting conditions and during postural and mental stress. Methodologically, multiscale analysis allowed us to focus on contributions to the information transfer which are related to specific oscillations with physiological meaning. In this work, the information transfer assessed for the original (raw) time series, which is typically computed in cardiovascular and cardiorespiratory analysis [22,26,27,28,31], is contrasted with that assessed specifically considering only slower (VLF and LF) oscillations. Moreover, the implementation of PID allows to highlight unique contributions of the two sources (RESP and SBP) to the target (RR) and to separate redundant from synergistic interactions between the sources. In this work, the unique information from SBP to RR reflects the strength of the high-pressure baroreflex influence on the HRV, while the unique information from RESP to RR reflects respiratory influences on HRV independent of the baroreflex. Regarding the interaction between sources, the analysis of redundancy and synergy seems to be particularly relevant in the evaluation of the cardiovascular control [6,38,39]. Here, the separation of redundancy and synergy components allowed by PID favours the evaluation of the importance of baroreflex-mediated and nonbaroreflex effects of respiration on HRV (RESP→SBP→RR and RESP→RR pathways, respectively). In particular, the use of PID is important to properly assess synergistic interactions between RESP and SBP, because the general prevalence of redundancy over synergy (negative interaction information) observed in the present and in previous studies [28,39] obscures the synergy amount in the IID.

### 4.1. Partial Information Decomposition of Cardiovascular and Cardiorespiratory Interactions During Postural and Mental Stress—Raw Data Analysis

Our results document that at the original time scale at which the time series are observed (τ = 1), a significant amount of information is transferred from respiration and systolic blood pressure towards the heart period in all the considered experimental conditions. The decomposition of this joint information transfer led us to evidence different types of contributions with physiological meaning. In the resting supine position, the unique information transfer from RESP to RR clearly prevails over the unique transfer from SBP to RR (Figure 4c,d), while redundancy and synergy between RESP and SBP are both present (Figure 4e,f) with a prevalence of redundancy over synergy (Figure 4b). These patterns of information transfer suggest that the two physiological mechanisms which may potentially give rise to RSA are simultaneously active: nonbaroreflex effects, possibly of central origin [37], are reflected by the high unique transfer from RESP to RR; baroreflex-mediated effects [40] are reflected by the prominent redundancy between RESP and SBP sending information to RR.

The postural stress induced by HUT was associated with a markedly higher joint information transfer from SBP and RESP to RR (Figure 4a), which resulted from a significant increase of the unique transfer from SBP to RR (Figure 4c) and of both the redundant and the synergistic transfer (Figure 4e,f); redundancy increased more than synergy because the interaction transfer was significantly lower (Figure 4b), while the unique transfer from RESP to RR decreased significantly (Figure 4d). The picture emerging from these results is that postural stress induces a strong activation of the baroreflex, related both to RR oscillations purely driven by SBP (unique transfer SBP→RR, presumably associated with LF oscillations) and to RR oscillations driven first by RESP and then by SBP (baroreflex-mediated RSA, path RESP→SBP→RR). On the other hand, the drop of the unique transfer from RESP to RR indicates a dampening of the nonbaroreflex path of RSA. These findings are in agreement with previous works reporting baroreflex activation and weakening of RSA with HUT [22,25,31] and complement them suggesting that RSA is dampened along the nonbaroreflex path, while is actually enhanced along the baroreflex one. An increased involvement of the baroreflex in the generation of RSA during orthostasis was also recently suggested in [23,25,41].

Contrary to the postural stress, the mental stress induced by MA did not produce significant alterations in the joint information transfer or in the redundant or synergistic transfer, nor in their balance (Figure 4a,b,e,f). On the other hand, MA evoked an increase of the unique transfer from SBP to RR and a simultaneous decrease of the unique transfer from RESP to RR similar to those observed during HUT (Figure 4c,d). These results suggest that the baroreflex-mediated RSA (RESP→SBP→RR) is not activated during mental stress like it is during postural stress, while nonbaroreflex RSA mechanisms (RESP→RR) are blunted by both postural and mental stress. The overall weakening of the influence of RESP on RR is in agreeent with vagal inhibition provoked by stress challenges [22,25,31]. During the MA task we also found significant increase in the breathing rate in comparison with previous supine recovery (*p* < 0.001), which—as was shown in a previous study—could also diminish the effect of RSA [42]. On the other hand, baroreflex effects unrelated to respiration (SBP→RR) are significantly larger during MA. We suggest that cortical mechanisms related to mental stress elicit vasomotor reactions reflected in SBP changes and subsequently transferred to HRV through the baroreflex.

It is worth noting that while PID information decomposition was applied in this study to healthy subjects, it may be useful also to reveal the alteration of the patterns of cardiovascular and cardiorespiratory interactions which occur in pathological conditions. Previous studies have demonstrated the link between multivariate information measures applied to physiological time series and pathophysiological conditions [3,17,18,30]. For instance, in a previous study we observed that a measure of causal coupling from SBP to RR related to transfer entropy increased significantly and was maintained at high levels during prolonged head-up tilt in healthy subjects, while it dropped to low values when assessed in patients experiencing neutrally-mediated syncope [17]. In this context, future extensions of the present work will be aimed at investigating the mechanisms related to postural syncope in terms of the altered unique, synergistic and redundant contributions brought to heart rate variability by the arterial pressure and respiration sources.

### 4.2. Multiscale Information Decomposition of Cardiovascular and Cardiorespiratory Interactions—Focus on Slower Oscillations

At rest, the unique TE from SBP to RR is the only index that is higher for slower oscilations compared to the raw data; in the multiscale representation one can appreciate its increase from τ ≥ 3 (Figure 3c) and comparing τ_1_ and τ_2_ in the scale-specific analysis (Figure 4c). We suggest that this result arises from the fact that SBP has the most relevant part of its dynamics in the LF and VLF bands, and therefore it may transfer more information to RR when assessed only for slower oscilations. This indicates that the baroreflex is an important mechanism for slower heart rate oscillations, and supports previous studies stressing the necessity to analyse the baroreflex in the LF band and evidencing that the interpretation of baroreflex coupling and gain can be less reliable when other mechanisms (e.g., nonbaroreflex mechanisms of RSA) are involved in the genesis of RR oscillations [43].

As discussed in Section 4.1, at the time scale of the original time series both postural stress and mental stress strengthened respiration-unrelated baroreflex effects (higher $U_{\mathrm{SBP}\to \mathrm{RR}}$) and weakened nonbaroreflex RSA (lower $U_{\mathrm{RESP}\to \mathrm{RR}}$). On the other hand, focusing on slower oscillations only, we noticed that such effects remained at the resting level or even showed opposite trends: at long time scales $U_{\mathrm{SBP}\to \mathrm{RR}}$ decreased significantly during MA (Figure 3c) and $U_{\mathrm{RESP}\to \mathrm{RR}}$ increased significantly during HUT (Figure 3d). These results are novel and unexpected and indicate that both baroreflex and respiration influences on HRV include complex multiscale patterns flexibly responding to stress challenges. This suggests that lower baroreflex-mediated transfer of SBP oscillations to HRV occurs during mental stress and that slowly varying respiration influences (mostly related to spontaneous changes of the respiratory pattern) are also transferred to slower HRV oscillations, and this transfer is stronger during postural stress. These results stress the importance of emplying a multiscale approach in the analysis of cardiovascular and cardiorespiratory interactions. However, we notice that the decrease of $U_{\mathrm{SBP}\to \mathrm{RR}}$ with MA and the increase of $U_{\mathrm{RESP}\to \mathrm{RR}}$ with HUT are not statistically significant when the analysis at long time scales is performed adjusting for the mean heart rate (i.e., computing τ_2_ individually for each subject; Figure 4c,d). Therefore, further studies are needed, probably considering longer time series, to support the consistency of these complex interaction patterns observed when only VLF and LF oscillations are analysed.

The multiscale patterns of redundancy and synergy depicted similar behaviors across a wide range of time scales, evidencing that both measures decrease with time scale, are significantly higher across several time scales during postural stress and do not change during mental stress (Figure 3e,f). The significantly higher redundancy and synergy found during HUT also considering slower oscillations only could be related to an involvement of respiration also in the LF band (probably as an effect of changes in the respiration pattern) as observed previously [44]. Such an involvement could also explain the shift from the prevalence of redundancy to the prevalence of synergy observed during HUT moving from scale 1 to scales 3, 4 and 5 (from negative to positive interaction transfer, Figure 3b). This behaviour, together with the increase of the unique transfer from RESP to RR at longer time scales (Figure 3d), suggests that the two pathways whereby respiration affects the heart rate (RESP→RR and RESP→SBP→RR) are actually more balanced with each other when only slow oscillations are observed.

## Figures and Tables

**Figure 1 entropy-21-00526-f001:**
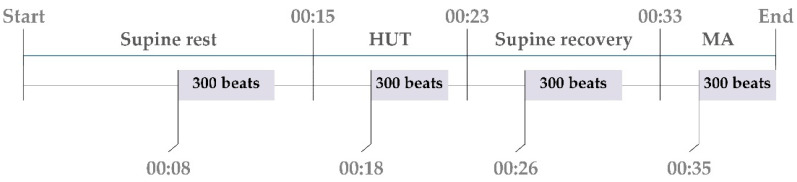
Timeline of the study protocol with indication of the timing of the sequences of 300 beats selected for the analysis.

**Figure 2 entropy-21-00526-f002:**
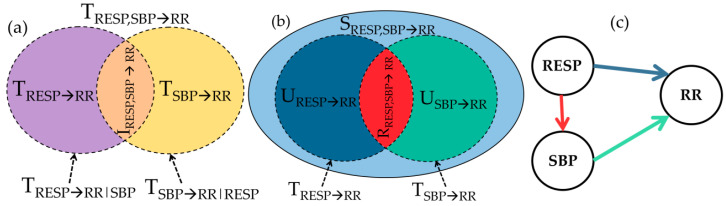
Information decomposition of cardiovascular and cardiorespiratory interactions. (**a**) Interaction information decomposition diagram depicting how the joint TE from respiration pattern (RESP) and systolic blood pressure (SBP) to RR is expanded as the sum of the two individual TEs from RESP to RR (violet) and from SBP to RR (yellow), plus the interaction TE from RESP and SBP to RR (orange); (**b**) Partial Information decomposition diagram showing how the joint TE from RESP and SBP to RR is expanded as the sum of the two unique TEs from RESP to RR (blue) and from SBP to RR (green), plus the redundant TE (red) and the synergistic TE (light blue) from RESP and SBP to RR; (**c**) Causal interaction diagram depicting the direct effects of RESP on RR (blue arrow), the effects of RESP on SBP (red arrow) and the effects of SBP on RR (green arrow).

**Figure 3 entropy-21-00526-f003:**
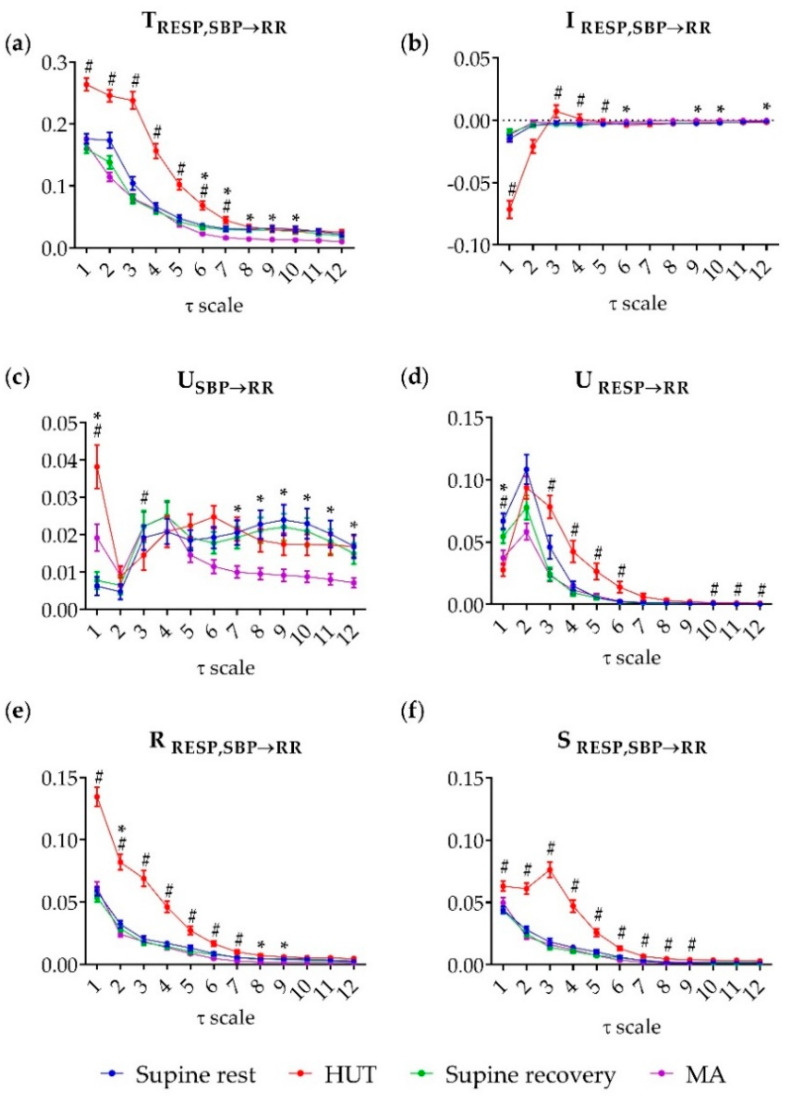
Multiscale information decomposition during the four phases of the experimental protocol (supine rest, HUT; supine recovery, MA). Plots represent the distributions (median and interquartile range) of (**a**) the joint TE (T_RESP,SBP→RR_), (**b**) the interaction TE (I_RESP,SBP→RR_), (**c**) the unique TE from SBP to RR (U_SBP__→__RR_) and (**d**) from RESP to RR (U_RESP→RR_), (**e**) the redundant TE (R_RESP,SBP→RR_) and (**f**) the synergistic TE (S_RESP,SBP→RR_), computed as a function of the time scale τ. # denotes statistically significant difference between the first and second phase (an effect of HUT) and * denotes statistically significant difference between the third and fourth phase (an effect of MA).

**Figure 4 entropy-21-00526-f004:**
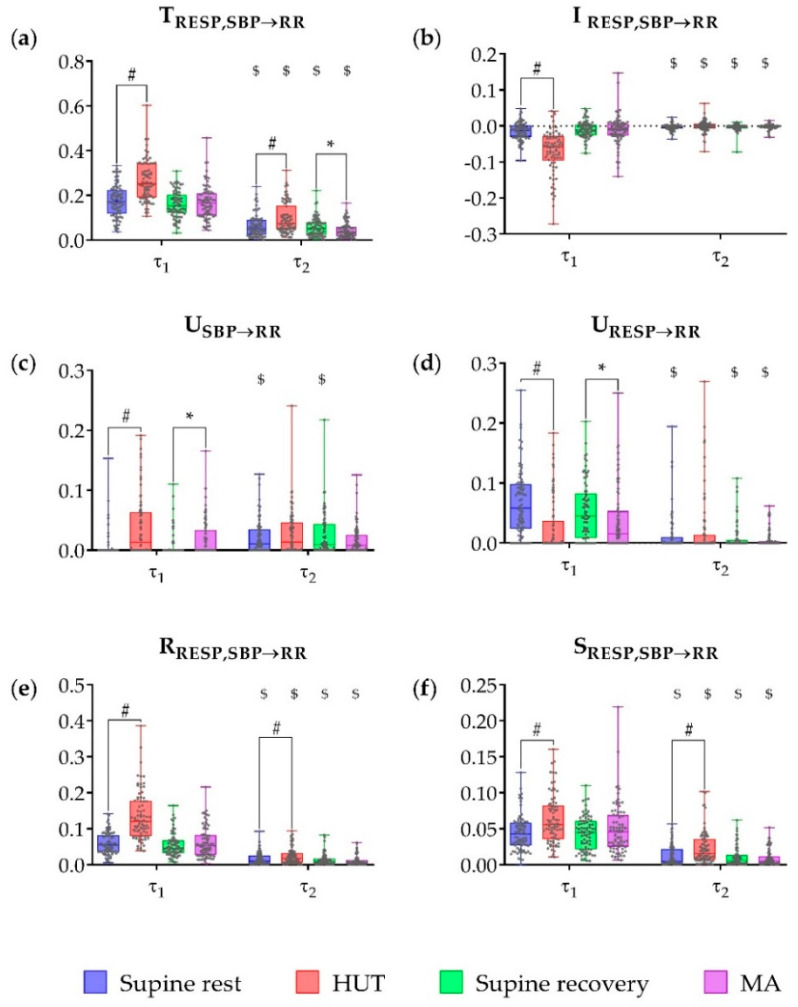
Multiscale information decomposition during the four phases of the experimental protocol (supine rest, HUT; supine recovery, MA) calculated for the scales representing raw data (τ_1_ = 1) and slower oscillations only (τ_2_). Plots depict the distributions (box plots and individual values) of (**a**) the joint TE (T_RESP,SBP→RR_), (**b**) the interaction TE (I_RESP,SBP→RR_), (**c**) the unique TE from SBP to RR (U_SBP→RR_), (**d**) the unique TE from RESP to RR (U_RESP→RR_), (**e**) the redundant TE (R_RESP,SBP→RR_) and (**f**) the synergistic TE (S_RESP,SBP→RR_). # Denotes statistically significant difference between the first and second phase (supine rest vs. HUT), *denotes statistically significant difference between the third and fourth phase (supine recovery vs. MA), and $ denotes statistically significant difference between raw data (τ_1_) and slower oscillations (τ_2_) during the same phase.

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
