# Peer review of "Multiscale Information Decomposition Dissects Control Mechanisms of Heart Rate Variability at Rest and During Physiological Stress"

_entropy, 2019, doi:10.3390/e21050526_

Round 1

Reviewer 1 Report

This is a high quality paper which applies the author's powerful and elegant multiscale information decomposition to an important empirical problem. 

I have only one relative small concern. In section 2.5 (statistical analysis) the authors do not spell out if post-hoc comparisons and in general statistical significance is corrected for multiple comparisons, and if so how. This appears relevant at least for the two post-hoc comparisons.

Author Response

Manuscript by Krohova et al. - Response to Reviewer 1

We would like to thank the reviewer for his/her accurate and constructive feedback on our paper. We have addressed all comments raised, and believe that the paper is improved and is more complete after the revision. Below we report the point-by-point replies to the reviewer comments also indicating the relevant changes made to the manuscript. The replies and the changes in the revised MS document are typed in red font.

Q: This is a high quality paper which applies the author's powerful and elegant multiscale information decomposition to an important empirical problem.
A:  We thank the reviewer for the evaluation of our paper, and for the positive comments.

Q:  I have only one relative small concern. In section 2.5 (statistical analysis) the authors do not spell out if post-hoc comparisons and in general statistical significance is corrected for multiple comparisons, and if so how. This appears relevant at least for the two post-hoc comparisons.
A: The Conover test was used as a post hoc test for the Friedman test, considering the need of correcting for multiple comparisons. This aspect was clarified in the revised paper in Sect. 2.5.

(See attached file.)

Reviewer 2 Report

The paper “Multiscale information decomposition dissects control mechanisms of heart rate variability at rest and during physiological stress” deals with the application of the joint information transfer from respiration and systolic blood pressure to beat-to-beat interval that is decomposed into different components called unique, redundant and synergistic. These components are analyzed in different time scales applying a multiscale analysis to data from a protocol involving different stress after each other as a head-up tilt test and mental arithmetics. The paper is well-written and organized, the theoretical aspects are well explained and the authors indicate specific literature references to the reader that complete the information about different techniques as the multiscale analysis. Regarding the results, these are very interesting and emphasize the advantages of the multiscale analysis and dissection of the joint transfer information. Just some minor elements that the reviewer would like to know the opinion of the authors:

1.     The paper just includes healthy subjects. It could be really interesting to analyze data from patients that suffer vasovagal syncope to see the behavior of the different components in comparison with healthy subjects. Could the authors speculate about the possible changes in the different components in the patient case?

2.     Regarding the time scale tau2 in equation 9, which is the dispersion of the tau2 value in healthy subjects? This may be important in the case of patients in order to ensure approximately the same time scale for the analysis of the different components.

3.     A minor typo on page 7 line 284, “sigificant.”

Author Response

Manuscript by Krohova et al. - Response to Reviewer 2

We would like to thank the reviewer for his/her accurate and constructive feedback on our paper. We have addressed all comments raised, and believe that the paper is improved and is more complete after the revision. Below we report the point-by-point replies to the reviewer comments also indicating the relevant changes made to the manuscript. The replies and the changes in the revised MS document are typed in red font.

Q1: The paper just includes healthy subjects. It could be really interesting to analyze data from patients that suffer vasovagal syncope to see the behavior of the different components in comparison with healthy subjects. Could the authors speculate about the possible changes in the different components in the patient case?
A1:This is an important point – the methods introduced in the current manuscript in healthy subjects are planned to be applied in future studies on patients with cardiovascular dysregulation. Considering the current knowledge about the interconnection inside this complex system it is very difficult to predict the results in a specific patient group. Previous studies indicate that a lower system complexity with a dominance of only several basic interconnections should be characteristic for patients with various impairments of cardiovascular control, including vasovagal syncope. Several studies indicate a baroreflex impairment in patients with vasovagal syncope and we suggest that these changes could be detectable by the information domain analysis introduced in this manuscript by the lower information transfer in SBP to RR direction. In the revised manuscript (Discussion, end of Sect. 4.1) we have added a paragraph where we outline this perspective referring to a previous work where altered causal information transfer from systolic pressure to heart rate was documented in the epochs preceding neurally-mediated syncope.

Q2:Regarding the time scale tau2 in equation 9, which is the dispersion of the tau2 value in healthy subjects? This may be important in the case of patients in order to ensure approximately the same time scale for the analysis of the different components.
A2:Thanks for the interesting comment and observation. The time scale tau2, which was calculated using Eq. 9 and rounded to the smallest possible integer value which is greater than or equal to the given number, are given in the following as median and range of values for the four phases of the protocol: during supine rest the time scale was 4 (3–5), during HUT 5 (4–7), during supine recovery 4 (3–6) and during MA it was 5 (3–7). These values were reported in the revised manuscript (Results, Section 3).

Q3:  A minor typo on page 7 line 284, “sigificant.”
A3: Thanks for noting the mistake, it was corrected.

(See attached file.)

Reviewer 3 Report

The authors perform a multiscale information decomposition of cardiovascular and cardiorespiratory interactions in a large group of healthy subjects recorded during relaxed physiological condition and during different types of physiological stress. The approach include RR, RESP and SBP recordings (non-invasive and short-term) to assess the two major physiological mechanisms underlying HRV and to enable an information-theoretic interpretation of these mechanisms depending on defined physiological conditions.

While this methods was introduced in before by the authors and applied in a preliminary study looking at very specific patients, I do appreciate the general concept of this follow-up paper to investigate all these information decomposition measures from the practical point of view by its application to a large group of subjects and during clearly defined physiological conditions. In my opinion, the performed differentiation of TE (joint, interaction, unique, redundant and synergistic) sounds encouraging in the field and will contribute to a further understanding of important physiological mechanisms.

The paper is (up to minor points) well-structured and -written. Theoretical background as well as methodological considerations including the coverage of related literature are given, technical / mathematical details of recordings, proposed methods and performed statistical evaluations are well described, results are specified, discussed and summarized.

Minor comments:

1) page 3 line 108- experimental protocol: Please remove the needless “for” before "(6 min)"!

2) page 3 – time series extraction: The description of the extraction of the four different segments (supine rest, HUT, supine recovery and MA – as stated before in section 2.1 and later on, e.g. in Figure 2) is kind of confusing. The authors should use the designation of defined phases at this point and not only state “first segment … started..., second segment…”. Possibly, also a figure with a simple time-scheme of extracted data would be helpful.

3) page 4 - figure 1: some of the designations used are difficult to read, please revise.

4) page 6 – lines 234 to 238 / line 245 to 249: the inline format of formulas the authors used is hard to read, please revise.

5) page 7 and 8 – figure 2 and 3: it would be better to arrange the subplots of figure 2 and 3 in the same manner ( (a) to (f) in both cases – one is 3 by 2 and one is 2 by 3).

6) page 8 - statistical evaluations in figure 3: The authors stated (page 6, line 259 to 260) to have also performed statistical tests between raw data (scale tau1) and slower oscillations (scale tau2) within the same phase (supine rest, HUT, supine recovery and MA). I do not really recognize any results of it in figure 3 (and possibly it would be too much to show in the same figure). The authors should think about an appropriate representation of these results of statistical evaluation.

Author Response

Manuscript by Krohova et al. - Response to Reviewer 3
We would like to thank the reviewer for his/her accurate and constructive feedback on our paper. We have addressed all comments raised, and believe that the paper is improved and is more complete after the revision. Below we report the point-by-point replies to the reviewer comments also indicating the relevant changes made to the manuscript. The replies and the changes in the revised MS document are typed in red font.

Q1: page 3 line 108- experimental protocol: Please remove the needless “for” before "(6 min)"!
A1: Thanks for noting the typo, it was corrected.

Q2: page 3 – time series extraction: The description of the extraction of the four different segments (supine rest, HUT, supine recovery and MA – as stated before in section 2.1 and later on, e.g. in Figure 2) is kind of confusing. The authors should use the designation of defined phases at this point and not only state “first segment … started..., second segment…”. Possibly, also a figure with a simple time-scheme of extracted data would be helpful.
A2: The sentence was rephrased to avoid confusion (see text in Section2.2). Moreover, as suggested by the reviewer, we have also added a figure with the time-scheme of extracted data (page 3, new Fig. 1).

Q3: page 4 - figure 1: some of the designations used are difficult to read, please revise.
A3: The figure (now figure 2) was revised increasing the size of the designations so that they are now easier to read.

Q4: page 6 – lines 234 to 238 / line 245 to 249: the inline format of formulas the authors used is hard to read, please revise.
A4: To improve clearness, we avoided presenting the formulas inline, but rather inserted them as two new equations (Eqs. 10 and 11), while keeping the format and size suggested by the template of the journal.

Q5: page 7 and 8 – figure 2 and 3: it would be better to arrange the subplots of figure 2 and 3 in the same manner ( (a) to (f) in both cases – one is 3 by 2 and one is 2 by 3).
A5: According to the suggestion of the reviewer, we have rearranged Figure 2 (now Figure 3) in a 3 by 2 format to adhere to the style of Figure 3 (now Figure 4).

Q6: page 8 - statistical evaluations in figure 3: The authors stated (page 6, line 259 to 260) to have also performed statistical tests between raw data (scale tau1) and slower oscillations (scale tau2) within the same phase (supine rest, HUT, supine recovery and MA). I do not really recognize any results of it in figure 3 (and possibly it would be too much to show in the same figure). The authors should think about an appropriate representation of these results of statistical evaluation.
A6: We have modified figure 3 (now Figure 4) to include results of all statistical tests, and have extended the text in the figure caption to clarify the description of all statistically significant results.

(See attached file.)
